# Implementing supportive supervision in acute humanitarian emergencies: Lessons learned from Afghanistan and Ukraine

Nadeen Abujaber[1]*, Meg Ryan[1], Kelly A. McBride[2], Pia Tingsted Blum[2], Michelle Engels[2], Anna Didenko[3], Hannah Green[4], Catia Sofia Peres de Matos[2], Shona Whitton[2], Frédérique Vallières[1]

1 Trinity Centre for Global Health, Trinity College Dublin, Dublin, Ireland, 2 Reference Centre for Psychosocial Support, International Federation of the Red Cross and Red Crescent Societies, Copenhagen, Denmark, 3 Rehabilitation and Support Department, Red Cross Society, Kyiv, Ukraine, 4 Mental Health and Psychosocial Services Department, Save the Children, Kyiv, Ukraine

* nabujabe@tcd.ie

**Data Availability Statement:** Data contains potentially identifying or sensitive participant information. It has been recommended by the

## Abstract

Mental Health and Psychosocial Support (MHPSS) practitioners working in humanitarian contexts are at significant risk of mental health conditions, ultimately hindering the quality and sustainability of their work. Supportive supervision has shown to be effective in improving the wellbeing of MHPSS staff and volunteers and enhancing the effectiveness of MHPSS service delivery. Despite these proven benefits, there is a lack of standardised guidelines to inform supportive supervision within humanitarian contexts. To address this gap, the Trinity Centre for Global Health and the International Federation of the Red Cross Red Crescent Societies' Reference Centre for Psychocosial Support co-developed the 'Integrated Model for Supervision' (IMS) Handbook and supporting tools and led IMS trainings with four humanitarian organisations in Ukraine, Afghanistan, Jordan, and Nigeria from June-August 2021. The subsequent acute humanitarian emergencies that occurred in Afghanistan and Ukraine provided the opportunity to (i) examine the implementation of the IMS in the acute stages of *two* humanitarian crises and (ii) identify the challenges and lessons learned from this process. This study employed a case study design using semi-structured qualitative interviews with five MHPSS personnel (female: 4; male: 1) who had received training in the IMS and were directly involved in the implementation of supportive supervision using IMS guidelines in either Ukraine or Afghanistan. Results showed that participants identified the key steps needed for the implementation of supportive supervision and reported two significant barriers to implementation including the stress of a humanitarian crisis leading to competing responsibilities and priorities, staff shortages and time constraints as well as the challenge of creating a new supervision structure when none had existed previously. Overall, participants felt that the IMS resulted in improved knowledge, confidence, perceived support, team cohesion, staff wellbeing and was a helpful blueprint to guide the implementation of supportive supervision in humanitarian contexts.

Research Ethics Committee to not make data publicly available. Data can be made available upon reasonable request by contacting the data protection office at Trinity College Dublin at the following email: dataprotection@tcd.ie.

**Funding:** This study is made possible by the support of the American People through the United States Agency for International Development (USAID; Grant #: 720FDA19IO00106 to FV). The contents of this study are the sole responsibility of the authors and do not necessarily reflect the views of USAID or the United States Government. The funders had no role in the study design, data collection and analysis, decision to publish, or preparation of the manuscript.

**Competing interests:** The authors have declared that no competing interests exist.

## Introduction

Mental health and psychosocial problems are commonly reported within conflict affected populations [1]. Specifically, a greater prevalence of post-traumatic stress disorder (PTSD), depression, and psychosis, among other severe forms of psychological distress, have been observed within humanitarian settings [2]. In a concerted effort to mitigate this observed risk, the integration of mental health and psychosocial support (MHPSS) programming is increasingly recommended within humanitarian responses [3].

Like other humanitarian responders, those tasked with the delivery of MHPSS are exposed to several stressors including harsh and sometimes dangerous working conditions, long working hours, and large caseloads. MHPSS practitioners are also at increased risk of vicarious trauma, and, in cases where MHPSS practitioners are also members of affected communities, vicarious trauma can be further compounded by personal trauma [4–6]. Consequently, MHPSS staff and volunteers are also at heightened risk of anxiety, depression and PTSD, which can negatively impact the quality and sustainability of their work [7]. Reasonable workloads, with adequate resourcing, appropriate training, and regular supportive supervision are recommended to ensure the effectiveness of MHPSS service delivery, while also safeguarding the safety and wellbeing of MHPSS staff and volunteers [8,9].

Supportive supervision, defined by McBride and Travers [10] as a 'safe, supportive, confidential and collaborative relationship between a supervisor and supervisee, where supervisees can voice their difficulties, discuss challenges and be recognised for their successes, receive constructive feedback and emotional support, and build their technical skills and capacity', is associated with greater staff motivation, enhanced wellbeing, increased knowledge, competency, and confidence, decreased burnout and turnover rates and improved quality of health services [11–13]. The primary factors encompassed within supportive supervision are professional development, intervention monitoring fidelity, and emotional support [14,15]. Despite the proven benefits of supportive supervision, supervision for health programming is often inconsistent, poorly resourced, and of low quality [16]. Furthermore, there is a lack of standardised guidelines to inform supportive supervision within humanitarian contexts. To address this gap, the 'Integrated Model for Supervision' (IMS) Handbook and accompanying materials were developed in 2021.

The IMS was created by a collaboration between the Trinity Centre for Global Health (TCGH) and the International Federation of the Red Cross, Red Crescent Societies' Reference Centre for Psychocosial Support (PS-Centre). It features standardised guidelines for incorporating and employing evidence-based supportive supervision practices for MHPSS practitioners operating in humanitarian contexts. The IMS was developed using participatory action research approaches spanning multiple stages of stakeholder consultation, including a desk review, regional workshops, key informant interviews [17], and Delphi techniques [18]. It was first piloted with four humanitarian organisations based in Ukraine, Afghanistan, Jordan, and Nigeria. Training on the IMS took place from June to August 2021 and involved a pre-training meeting, four training modules, and follow-up supervisory support and implementation consultations, delivered in line with the apprenticeship training model [19]. Within Module 1, attendees from leadership/management focused on background information on the IMS and why supervision is considered critical to MHPSS programming. As part of Module 2, supervisors/future supervisors were trained in types of supervision, how to structure sessions, ethics and boundaries, as well as practical skills such as self-care, reflective practice, effective facilitation, and building alliance. Supervisees attending Module 3 explored how to make the most of their supervision experience, and practical tools such as how to do case presentations and use role play effectively. The final module, Module 4, combined attendees once more to explore

the implementation of the IMS, cultural adaptation, monitoring and evaluation, and next steps. All IMS Trainings were conducted in English and were held online due to the COVID-19 pandemic restrictions. Follow-up supervision (group and individual) and 1-on-1 implementation support was offered to all organisations who took part.

Unfortunately, the situations in both Afghanistan and Ukraine deteriorated shortly after the IMS training, with the events of 15th of August 2021 and February 24th 2022, respectively, changing the course of each country. The acute humanitarian emergencies that followed provided the undesired opportunity to examine the appropriateness, acceptability and feasibility of implementing the IMS within these two humanitarian contexts. The current study therefore aims to understand *'what happened after the IMS training?'* by describing the (i) process of planning for and implementing the IMS post-training as well as in the acute stages of *two* humanitarian crises and the (ii) identification of challenges and lessons learned from the IMS' implementation in both Afghanistan and Ukraine, with the goal of further developing and strengthening the IMS for future use within other humanitarian contexts.

## Methods

### Study design, procedures and participants

The current study employed a case study design. Participants were recruited between February and March 2022 through purposive sampling and included five MHPSS personnel (female = 4; male = 1) who had received training in the IMS and were directly involved in the implementation of supportive supervision using IMS guidelines in either Ukraine or Afghanistan. Two participants out of the total 16 IMS training participants from Afghanistan were employed by a non-governmental organisation (NGO) working in Afghanistan and three participants out of the total 14 IMS training participants from Ukraine were employed by an NGO working in Ukraine. Two of the participants held management positions within their organisation and the other three participants were supervisors. Two of the participants interviewed twice; once to discuss their initial implementation plan one month after the initial pilot training, followed by a six-month follow-up interview. Participant details can be found in Table 1, with the participants who were interviewed twice denoted in bold text.

### Humanitarian contexts: Afghanistan and Ukraine

While Afghanistan has experienced decades of insecurity and conflict, leaving millions of people displaced and living in poverty, the humanitarian crisis escalated after August 15th, 2021. The country is now experiencing a grave humanitarian crisis and international aid has largely been cut off, depriving people of necessities such as food and medicine [20]. According to the United Nations, 95 percent of Afghan people are now going hungry [20]. Health services in Afghanistan have also gotten worse, with many of the NGOs who had previously acted as the

**Table 1. Participant details.**

| Participant Code | Gender | Role | Years of experience in MHPSS | Years providing supervision | Years receiving supervision | Context |
|---|---|---|---|---|---|---|
| **A** | **F** | **Management** | 8 | 2 | 8 | **Afghanistan** |
| B | M | Supervisor | 2 | 8 | 0 | Afghanistan |
| C | F | Supervisor | 3 | 3 | 0 | Ukraine |
| D | F | Supervisor | 3 | 3 | 2 | Ukraine |
| **E** | **F** | **Management** | Not Available | Not Available | Not Available | **Ukraine** |

F: Female, M: Male.

primary health service delivery mechanism forced to withdraw from the country [21]. Many Afghan workers have been forced to leave their jobs due to low wages or, in the case of female staff, are prevented from working altogether. Despite a high prevalence of psychological distress among the population [22], with trauma and exposure to violence acting as key determinants of mental health outcomes [23,24], mental health resources in Afghanistan remain scarce [25].

In Ukraine, over 13.4 million people have been displaced since Russia deployed its military into the country on February 24th, 2022. More than 23,300 civilian casualties have been recorded in the country, including over 8,700 deaths [26]. Russian attacks have also destroyed much of the infrastructure in the country, leaving thousands of Ukrainians without basic necessities, and unable to access medical assistance [27]. As a country with a recent history of conflict and unrest, previous studies have highlighted the negative impact on the mental health of young people [28]. Increased levels of PTSD, anxiety and depression have also been found due to the trauma experienced as a result of war, such as exposure to violence and abuse, and the loss of family and friends [28]. Altogether, there is a need to provide proactive and preventative MHPSS services to the population of Ukraine, and to ensure that MHPSS staff are supported in this task.

## Data collection

For security reasons and given the difficulty in obtaining in-country ethical approval in the setting of acute humanitarian crises, interviews were conducted online via Zoom [29]. All interviews were audio-recorded after receiving permission from the participants and were stored on a secure computer, accessible only to the research team, in a password-protected file. Audio files were deleted permanently once transcription was completed. All participant data was encrypted and stored on OneDrive, accessible only to research team during the study. Interviews lasted between 30–60 minutes and were conducted by the first (NA) or second (MR) authors in English. Semi-structured interviews were chosen because they allow the delineation of key topics but also provide enough flexibility for the open-ended exploration of ideas and themes [30]. Questions explored the impact of the humanitarian emergencies on the participants and their work, the role of supportive supervision and the IMS during humanitarian emergencies, whether the IMS was appropriate, acceptable and feasible to implement during acute humanitarian contexts and discussing the facilitators and barriers to implementation as well as lessons learned from the implementation of the IMS in these acute phases of an emergency.

## Data analysis

Interviews were transcribed verbatim in English. Data was anonymised at the point of transcription with participants being represented by unique codes. Thematic analysis using Braun and Clark's [31] six-phase framework was applied to the interview transcripts in the following manner. First, the transcripts were compared to the audio-recordings to ensure accuracy and data familiarisation. Second, each transcript was examined systematically for code identification. Codes were then compared to each other to find overlapping ideas and create subthemes. Fourth, additional subthemes were elicited by re-evaluation of each transcript. Fifth, each subtheme was defined with a clear descriptive name. Finally, direct quotations were included to support each subtheme. Subthemes were joined and grouped into themes.

**Ethics statement.**   This study received ethical approval from Trinity College Dublin Health Policy and Management/Centre for Global Health Ethics Committee on January 17th,

2020. All participants provided written consent in English prior to their interviews. Participants were asked for approval for the use of their anonymised quotes prior to publication.

## Results

The themes and subthemes from the thematic analysis of qualitative interviews from the five participants in this study are summarised in Table 2. Participants identified the steps of developing and planning the implementation of supportive supervision per the IMS method, the barriers faced during this process, and their initial impressions of implementing the IMS supportive supervision in the acute phases of a humanitarian emergency.

### Planning for implementation

Participants discussed during their interviews the particular challenge of implementing supervisory structures within their organisations when none had previously existed, highlighting that they were 'starting from scratch', often wondering how to start, when to start, with whom to start and with what resources. That said, participants identified key steps and processes in their implementation planning, including a needs analysis, human resource allocation, contextualisation, advocacy, monitoring and evaluation, dissemination and scale up, and sustainability measures.

**I. Needs analysis.** As an initial step, participants emphasised the importance of evaluating and understanding the needs of their team members directly in order to create the most appropriate and effective supervision structure. The two main aspects of this needs analysis identified by Participant B were *exploring expectations*, or the importance of evaluating the expectations about supervision from all members of their team, especially supervisees, and to use these expectations as part of the implementation plan, and *identifying gaps* in the current organisational structure to ensure that the implementation plan addresses them directly:

**Table 2. Themes and subthemes for implementation planning, barriers to implementation and review of the IMS.**

| Themes | Subthemes |
|---|---|
| *Planning for implementation* | I. Needs analysis |
| | II. Human resource allocation |
| | III. Contextualisation |
| | IV. Advocacy |
| | V. Monitoring and evaluation |
| | VI. Dissemination & scale up |
| | VII. Sustainability measures |
| *Barriers to implementation* | I. Impact from humanitarian emergencies |
| | II. Lack of existing supervisory structure |
| *Initial impressions from implementation* | I. Access to knowledge |
| | II. Confidence and perceived support |
| | III. Team cohesion |
| | IV. Wellbeing |
| | V. Helpful tool |
| | VI. Blueprint for best practice |
| | VII. Applicability cross-sectors |
| | VIII. Practical guides needed |

*'I see it generally as the way the objective of IMS, what the expectation of our supervisees will have from our end so in that case definitely it is useful to work with our team what could be their expectation, and based on their expectation, we need to set goals.'*

*'One of the key components [in our implementation plan] is emotional support because this is something missing when we were having the supervisory role in our organisation.'*

**II. Human resource allocation.** Most of the participants highlighted the challenge of having a limited number of qualified staff knowledgeable in implementing supportive supervision. As Participant E described, '*to finalise and adapt these materials and IMS tool, from our side, we just need additional human resources, people who will be able to spend more time sitting together with me, developing the whole supervision system.*' Participants thus emphasised the importance of building human resource capacity in their implementation plans by using experienced staff, both internal and external to their organisation. As Participant A explained, '*When there's only two of you, I'm really wanting to just try and think how I can use the people that were in the previous training, and how I can use [the IMS trainers] as I create the implementation plan.*' Although they acknowledged that all staff would benefit from the IMS training, deciding '*who do we prioritise as needing this training*' was a concern given the financial constraints limiting the number of staff that could be trained in supportive supervision practices and the need to account for *high staff turnover* in their organisations. As stated by Participant A: 'we need to do as many people who can train or who can facilitate as possible or else, soon as I leave, it kind of will die.' In addition, Participant E emphasised that those they select to train in supervision '*need to be paid and we actually need these people for supervision*', pointing to the need for protected roles as supervisors.

**III. Contextualisation.** Many participants emphasised the importance of adapting the IMS implementation plan to make it as acceptable and appropriate to their context as possible through *translation* and *organisational and contextual adaptation*. For many, this meant the need to translate the IMS Handbook and resources from English to the primary language of their context to improve their staff's understanding of the supervisory guidelines and to increase their staff's contribution to the implementation of supervision in their organisation. As reported by Participant C: '*It would be great to make a translation from the Handbook in Ukrainian, it will be easier to ask the colleagues to read it and try to have a discussion with colleagues who don't know English about the supervision and maybe the ideas of these colleagues will be helpful for us like using supervision in everyday work.*'

Participants also noted the important process of organisational and contextual cultural adaptation of the IMS. As described by Participant E: '*It is not a book that you can just take and use in the original version because information needs to be adapted to the specific context and for each organisation.*' In addition to the cultural adaptation of the guidelines themselves, Participant A discussed in their interview the importance of culturally adapting the supervisory training sessions as well: '*Sometimes with trainers within our context, they'll take things very literally so if in the manual it says you need to do two days with leadership they'll be like okay two days when that's already been done a year ago and you don't need to re hash it.*'

To this end, both participants A and E made attempts to include elements of supervision into existing trainings and structures as preliminary steps to IMS implementation. Participant A reported that during a routine training session for counsellors: '*I took that as an opportunity to introduce them to what is supportive supervision because many of them are supervisors. So I took that beginning intro session [From IMS Handbook] and added it to the training. So that when I say we are doing this type of training, they are already familiar with the ideas.*' She noted

that not only would this provide them with initial knowledge of supervision that could be built upon in future trainings, she was also able to identify staff particularly interested in supportive supervision who could serve as future advocates for implementation activities. Similar to Participant A's approach, Participant E advocated for adding elements of supportive supervision, extracted from the IMS Handbook, to their existing organisational structure because she did not think that '*this supervision mechanism can be or should be a separate system or structure.*' Her proposed solution was as follows:

> '*Maybe doing supervision not as supervision but each volunteer when they conduct their regular volunteer meetings, they can add to these meetings to make it more supportive. For example, You can ask them: How do they feel? Do they need additional support? Or do they want to discuss a specific case for example?. . .If we can use the knowledge [from the IMS] to conduct group meetings better, to provide a person who will be a supervisor with skills on how to better manage hard emotions, how to better support their peers, that will be enough.*'

**IV. Advocacy.** Participants also discussed the need for evidence-based advocacy measures within their organisations to highlight the *importance of supervision for all staff members*. All participants emphasised the importance of having a supervision structure in place, especially in chaotic and stressful humanitarian contexts. As described by Participant A: '*this is our advocacy point is saying in humanitarian contexts when you have no time, this is when supervision is actually so critical.*' Participant D felt that supervision is so critical that it should be '*an obligatory standard*', embedded within the organisational workflows and procedures rather than an elective process. While acknowledging the high initial costs associated with implementing supervision, Participant D also felt that the benefits of improved capacity and staff retention would be economical in the long term:

> '*We are not losing money with supervision but we are saving money because let's say we train 10 people and the cost of the ToT [training of trainers] is high but in the end, if we are not supporting them, we will only have 1 or 2 left and then again we will need to hire more people and then provide another ToT and this will cost more than before. They [management] also like to save money.*'

All participants reported that *advocacy measures* should be focused on management and leadership, citing the importance of supervision for all staff members within every department and at all levels of the organisation. As Participant A described, '*we also just have to have a few more of those leadership meetings to get a few more people on board understanding, even the line manager, doesn't see how it benefits the team because at the moment, I think they think it's benefitting me so actually to show it's for everyone.*'

Participant A also noted that time was also needed to align management's vision of supervision to that advocated by the IMS: '*we have scheduled to sit and discuss what is their vision with supervision because I think it is a bit different to mine. I think it is going to be more of that kind of checklist supervision where IMS is not about that. And figuring out, to get buy in, how this supportive supervision is actually needed and different to what they had envisioned but does feed into it. It's not separate but it can just strengthen it.*' To ensure leadership commitment, she reported that much time was spent having '*1-on-1 conversations with key members of our staff that would need to support it. Support it meaning giving their staff the time for training as well as the money for the training.*'

Many participants highlighted that the advocacy measures presented to management and donors needed the *availability of data* demonstrating the benefits of supervision. As stated by

Participant A: '*I really need to think through what we want to be measuring, because I think for me that's really critical so that I can prove its, not that I can say its 100% supervision but to show some sort of evidence.*' Once evidence has been amassed in support of supervision, participants gave recommendations for the best way to deliver this information to management and donors. Many participants suggested concise and comprehensive promotional information such as '*one page like an elevator pitch*' (Participant A) or a '*less than one-minute video*' (Participant B). Participant E recommended that this advocacy document should mention '*what is supervision, how it will be applied, why it is important, and if you have any results to add to that and show them how it will look like.*' Participant D suggested similar content with an emphasis on '*very simple language and I highlighted outcomes, what they will have in the end*? *Why need to put it [supervision]*? *They [management] understand the most when I talk to them with numbers.*'

Participants also mentioned that *receiving follow up support from technical experts* in supervision was not only instrumental in their implementation activities but could also be an important advocacy tool. For example, Participants B and E mentioned the presence of IMS technical team at advocacy meetings with management to help facilitate discussions and provide expert information and data.

**V. Monitoring and evaluation.** Participants emphasised the importance of monitoring and evaluation tools with *indicators* chosen to demonstrate the impact of their supervision system. Participants mentioned several indicators that they believe would be impacted by instituting supportive supervision in their organisations such as '*the level of skills and level of personal psychological wellbeing of our volunteers*' (Participant C) and '*wellbeing, motivation, skill building and feedback mechanism*' (Participant D). By checking '*baseline and then after supervision, some end line to see results if this is helpful for them*' (participant E), this process would help to answer questions such as '*do they feel better with supervision*? *Are they ready to go again and again to be a volunteer*? *Do they need better support*? *What kind of support can we provide to them to make them feel better as a volunteer*?' *(Participant D)*. Not only would this build evidence in support of supervision but could also be used internally to track the professional development and emotional status of their staff, allowing an individualised approach and thus, '*catering supervision to fit the needs of the supervisees*' (Participant D).

**VI. Dissemination plan and scale up.** Participants reported that since they were creating a new supervision structure within their organisations, they would need to *start small*, prioritising those most affected by the humanitarian crisis to 'make the IMS system work for an emergency response' (Participant E) and *scale up slowly* over time to reach best practice. All participants mentioned that their initial implementation plan was considered a pilot project within their respective organisations, with the potential for future expansion after advocacy measures and proof of benefit.

> '*In terms of scaling up quickly, which I suppose they want to do, but I can't do that. We are starting with a very small group and then hopefully with advocacy and showing staff feeling more supported, we can do more stuff either later this year or early next year but then kind of after that point you want to start moving and making it bigger. . . we already talked about with them about how to progress to best practice.*' (Participant A)

**VII. Sustainability.** After the initial pilot training, all participants described their ultimate goal of *strengthening the capacity* of their supervision system to ensure *sustainability*. All participants acknowledged that the supervision system cannot be sustained in the long term with only a few people trained in supervision. As stated by Participant A: '*I'm the one who will push it, but my lifespan with this office is not forever.*' Participant B felt that the best way to achieve

this was in a cascade fashion via '*the apprenticeship model and based on that, we can also build the capacity of our front-line staff.*' Participant E also felt that this method of training would be the most expedient and effective in a humanitarian emergency: '*I hope that with this training of trainers, we will be able to, in the following months, prepare more supervisors and more how supervision will be working for the big demand of all the volunteers and instructors involved in the emergency responses.*'

### Barriers to implementation

Participants highlighted multiple barriers affecting the implementation of supportive supervision, most prominent the *direct impact of humanitarian emergencies* including the redirection of staff and resources towards other forms of assistance, high staff turnover, insufficient time, and the *lack of an existing supervisory structure* in their organisations.

**I. Impact of humanitarian emergencies.** Participants discussed the deleterious effects that humanitarian emergencies have on MHPSS activities, organisational priorities and support, staffing and protected time for supportive supervision. As stated by Participant B: '*the country was in a coup, and our operation was also suspended due to the conflict and the changes of regime for months.*' Participants reported that when operations restarted, the focus was often not on supervision and MHPSS, but on emergency response. '*I think it changed because we had originally planned to develop the MHPSS activities but in the last few months, I have been working in the provision of emergency response and react to acute needs. It has not been in the area of development at all. My work has been meeting the basic needs of people with a huge scale of people that need to be supported*' (Participant E). Participants also discussed in their interviews the shift in organisational priorities during an acute humanitarian crisis as stated by participant E: '*In terms of supervision, what we have now and what we planned, we haven't implemented it at all. . . . And MHPSS is not a first priority in emergency. . . I just think in an emergency context, supervision is not a priority.*'

Moreover, staff shortages present prior to the acute phases of an emergency were even more prominent during the early phases of each crisis.

> '*I mean we've also had a large staff turnover, so like the one person that was trained that was quite strategic because they needed to be doing supervision for counsellors, he left*' (Participant A)

> '*When we have humanitarian responses and other projects, we have a limited percentage of the project allocated towards staff so usually one person does a lot. . . . you will not have additional resources to also have a supervisor to provide supervision for volunteers*' (Participant E)

During a humanitarian crisis, humanitarian practitioners are often inundated with multiple competing demands on their time, '*rushing to implement activities and provide support*' (Participant B). These competing tasks and excessive workloads resulted in the delayed implementation of supportive supervision per participant A. A self-described '*supervision fan*' who '*thinks it [supervision] is a* priority' and that the '*IMS has given us every resource we need*', she noted that:

> '*I think that is something within a humanitarian context that makes it difficult to carve out time and so often it gets bumped because when staff do not even know how to implement even a basic activity, it has to become a priority over supervision . . . I think that has been one of the biggest challenges: finding time to fit everything. You know, we are just juggling the whole time.*'

**II. Lack of a pre-existing supervision system.**   Prior to the IMS training and implementation, the two organisations from Afghanistan and Ukraine did not have a structured or standardised supervision process in place. Participants reported limited organisational understanding of and support for supervision, which they believe led to the under-prioritisation of resources needed for supervision, hindering effective implementation. *'After seeing all the other things that need to be done, I'm like is that actually a realistic outcome? Because part of me is like I'm so desperate to get this [supervision system] up and get people going but then am I setting myself up for failure? I don't want to start without having all of these things already in place' (Participant A)*. In addition, participants described the resistance from management and leadership to implement supportive supervision. As stated by Participant E: *'Sometimes, it is hard to explain, for example, why we need supervision? Because they don't understand what supervision looks like. They [management] don't have that clear picture and even if I were to try to provide them with a general understanding, it does not work 100% as we expect it to.'* Consequently, participants reported that supervision was often inconsistent and primarily focused on managerial tasks, rather than the emotional support or professional development of staff, with a lack of continuity in terms of who facilitated supervision sessions.

Participants also remarked that the process of implementing IMS supportive supervision was time intensive. The additional time needed for the follow up support sessions with technical experts to guide implementation activities made it challenging to attend these sessions consistently in the peak of the humanitarian crisis. Participants felt these sessions were an important resource and expressed their appreciation for the flexibility and availability offered by the technical experts in meeting their needs (for example, switching to text-message check-ins, changing times, etc.). Participants also noted that protected time was needed for training to strengthen capacity and additional time was needed post-training to build supervision into the workflows, which was also difficult within humanitarian contexts. Participant A remarked that time was needed to hire and train staff who would be responsible for rolling out the supervision system. Participant E agreed: *'If we will change the person who is responsible for the development of the whole IMS system, then this process will need to be first because we only hired the new person 2 days ago who will be more involved in establishing the supervision system.'* Furthermore, *s*taff who were supposed to be involved in implementing supervision were also overwhelmed with large workloads and competing responsibilities during the height of a humanitarian emergency, causing delays in implementation.

> *'The biggest barrier that can be in writing any training programs because in my unit there is only three persons and we have a lot of volunteers and you have a lot of different tasks so if I had the possibility to just be sitting and writing the supervision manual and writing the new training I would be very happy but for now, unfortunately, no possibility for this I would say.' (Participant C)*

Without a precedent for supervision in their organisation, participant E noted that it was difficult to convince staff, already overwhelmed, overloaded and exhausted, to participate in supervision sessions, noting '*we do not have any time and people are still working and are still able to do their work, manage everything with low resources without supervisions. . . They do not have time for rest and for sure, they do not want to spend additional time for supervision. . .. . . Maybe it is easier to explain to people what is the impact and why supervision is needed when people have less work, less stress for example.'* This experience was shared by participant A who felt that the time constraints and excessive workloads resulted in a lack of protected time to perform supervision effectively: '*It's all related to time because supervision, the practice all takes*

*time and so again, it's always in a humanitarian context like we don't have time, I'm trying to see how to make it a priority.'*

## Impressions of supervision implementation

Participants also reflected on their efforts to implement supervision within their respective organisations, with noted impacts on staff knowledge, confidence, perceived support, team dynamics and wellbeing. They also evaluated the IMS Handbook in terms of its effectiveness, acceptability and applicability, offering key suggestions for its improvement.

**I. Improved access to knowledge.** Participants reported that after implementing their supervision pilots using information from the IMS Handbook, their staff had improved access to knowledge about supervision.

*'The Handbook is very clear and the definition of supervision and different types of supervision is very good for the general understanding of what supervision should look like, for what purpose, what steps and questions we should make with our supervisees.' (Participant E)*

**II. Improved confidence and perceived supervision.** Participants reported the improvement in their staff's perceived confidence in conducting supervision and noted receiving more support from their own supervisor. As stated by Participant A:

*'When I spoke to them about supervision, they felt that personally, one: it had made them feel more capable. . . they were meeting with the counsellors at least once a month, they went and did some live supervisions and gave feedback, so I feel like, even though it had been short, they had implemented it and were feeling a lot more confident in providing it.'*

*'They said they felt more supported from their* [own] *supervisor because that supervisor was in the training as well.'*

**III. Enhanced team cohesion.** Participants also described in their interviews a cultural shift within the teams exposed to supportive supervision with positive team dynamics emerging.

*'They might not even realise that its due to the IMS but because of the IMS strategy of training, it's quite a reflective process. . . they are incorporating components of aspects within their team to create a sense of unity, a sense of like I can ask for help when I need it not just this sense of I'm the boss and you listen to me.'* (Participant A)

**IV. Focus on staff wellbeing.** Participants described the focus on providing emotional support and enhancing staff wellbeing. As stated by Participant B: *'In one suboffice, you can see that for the wellbeing of their staff, they have a space for having social activities even within the office, there are playgrounds for the staff when they are having some stress, they can come together and play for their wellbeing.'*

**V. Helpful tool for advocacy.** All participants noted that the IMS Handbook was a helpful tool to advocate for supervision within their organisations. Participant D stated that: *'I used the IMS to get the definitions: What is supervision? How it can be provided? What are the types of supervision? I used it for the background, to define the different types of supervision, for how long? How many people? Etc. So, I used it A LOT.'* Likewise, Participants A and E reported using the IMS Handbook to advocate for increased human resources for supervision.

*'I already have used the Handbook and extracted quotes for proposals and things even in terms of advocating for staff to be like 'the global standard says', so I think that in itself has been really helpful.' (Participant A)*

*'It [IMS] is something to help them understand how many staff are needed because usually it is done by volunteers but it can't just be done and controlled by volunteers, you need staff who will be responsible to conduct supervision.' (Participant E)*

**VI. Blueprint for implementation.** Participants discussed the role of the IMS Handbook as a framework for implementing supervision within their organisations, with the hopes of achieving best practice. As reported by Participant A: '*It's quite good in that it [IMS Handbook] doesn't just throw information at you and say: 'off you go and figure it out yourself'. It gives you time to think on how are you going to implement it.'* Participant D confirmed that the IMS Handbook establishes the standard for how supportive supervision should be done: '*I really appreciate you starting this process and I don't need to tell anyone anymore to believe in supervision. And now I will say: 'look, this is the standard of our work, so no excuses.'*

**VII. Applicability across sectors.** Participants A and B remarked that the benefit of the IMS Handbook went beyond the MHPSS sector and was applicable in other sectors within humanitarian contexts. Participant A reported that, in addition to MHPSS unit, their child protection team and '*even the health and nutrition team have added it [supervision] to their organogram, so it is there, it is in place, which is a massive achievement.'*

**VIII. Suggestions for the IMS handbook.** Several participants suggested the inclusion of practical examples and guidelines, especially regarding emotional support and self-care, because this was identified as a significant gap in organisations prior to the implementation of supervision.

*'Which type of emotional practices would be the best option based on our context? Whether it could be giving more space maybe after like having an emergency response and maybe some time to relax before coming back to the office, but this is something there is a gap and how we can propose some emotional support exercises that are good for the wellbeing of our own staff?' (Participant B)*

*'I am just thinking about the IMS Handbook and it is not clear how to work with hard emotions. And especially during an emergency response, if for example, you are speaking and dealing with people who have seen somebody die or was affected by some, I don't know, explosion... So, it is not clear how to calm this person and what to do for example... And also, what to do for this supervisor who is providing this kind of supervision, after the supervision is ended, they have to calm down, self-care for the supervisor.' (Participant E)*

## Discussion

This study aimed to describe the process of implementing supportive supervision using the IMS method within two acute humanitarian emergencies, identifying barriers to implementation and extracting pertinent lessons to apply to future iterations of the IMS. Consistent with the literature [4–6,8,16], participants highlighted the significant need and desire for supportive supervision given the overwhelming stressors and workloads experienced in humanitarian contexts. However, the speed of implementation was hampered by the lack of a pre-existing supervisory structure within their organisations. Participants delineated the necessary steps to prepare for a successful implementation: identify gaps, clarify expectations, prepare and train human resources, contextualise guidelines to both organisations and culture, advocate for

supervision using evidence gathered from monitoring and evaluation, followed by dissemination with conditions needed for scale up and sustainability. Given the chaotic nature of humanitarian contexts where time and resources are limited and priorities shift rapidly, participants advocated for flexibility, adjusting to the needs of the context and offered solutions such as starting small and integrating elements of supervision into the pre-existing system with the aim to ultimately develop a strong supervisory structure. These findings make an important contribution to the literature given the current paucity of data describing *process* considerations for integrating supportive supervision during acute humanitarian emergencies. The IMS Handbook [10] was created to fill this gap and provides important information on how to work towards best practice in supportive supervision within an organization and these detailed steps for implementation, as described by participants, serve as a practical addition to the IMS Handbook, applicable to diverse humanitarian contexts.

Results also highlighted advocacy as central to the success of implementing supportive supervision. This is consistent with Ryan et al. [32] who found that inconsistent or poor implementation of supportive supervision within an organisation was directly impacted by the poor understanding of and commitment to supportive supervision by management and leadership. Efforts should be made to promote and demonstrate the benefits of implementing supportive supervision, using some of the suggested mechanisms identified by participants such as educational videos, pamphlets and technical experts as sources of information and support. Which of these method(s) is optimal to achieve organisational buy-in for supportive supervision, however, remains unknown.

Consistent with the inclusion of emotional support as one of the three main priorities for supportive supervision within the IMS Handbook [10], participants underscored the significant need for emotional support during humanitarian contexts given the stressful and overwhelming work of MHPSS in fragile states. Also reflected in previous studies [e.g. 6,33], this finding is particularly important given the documented risk for MHPSS practitioners of developing mental health conditions and secondary traumatisation [7]. Even when supportive supervision could not be fully implemented, participants described efforts made to incorporate aspects of emotional support for their staff in various ways from raising awareness about the psychological impacts of humanitarian contexts on humanitarian MHPSS practitioners, to providing direct support and offering additional resources for follow up support. Given the high unmet needs and limited resources within humanitarian emergencies, identifying the most effective delivery format for emotional support within humanitarian contexts would be an important area for future research.

In addition to creating true advocates for supportive supervision as evidenced by participant D's belief that it should be an 'obligatory standard', results highlighted several benefits of the IMS supportive supervision system including increased access to knowledge, improved confidence and perceived supervision, and increased wellbeing; all of which have been previously noted as advantages of supportive supervision [6,11–13]. Though the literature documents the positive impact of supportive supervision on the quality and sustainability of MHPSS services provided by staff who receive supportive supervision, more longitudinal data is warranted.

## Practical implications

It is the hope that the findings of this study will not only be used by partner organisations to optimise their supportive supervision practices but to also encourage other organisations in diverse humanitarian contexts to recognise the importance of supportive supervision and prioritise its inclusion into organisational structures, using lessons learned from participants

in this study and accessing the freely available IMS Handbook and supplemental materials as guidelines to move towards best practice.

## Limitations and considerations for future research

This study is not without limitations. The study has a small sample size and may not represent the perspectives of all who participated in the IMS training and implementation. However, recruitment during a humanitarian emergency is challenging and every effort was made to recruit from the core group primarily involved in implementation in each context. In addition, there were more female than male participants and results may therefore not account for differences in the experiences and perspectives of each gender. Furthermore, none of the participants were supervisees, as a perspective key to capture in future studies examining a newly implemented supervision structure. Another limitation is that only two participants were able to give follow-up feedback six months into the implementation of the IMS supportive supervision. Given the time it takes to implement a new system, especially in the chaotic environment of humanitarian emergencies, it would be beneficial to have a more longitudinal examination. Finally, interviews were conducted in English and though participants were fluent in this language, for many, English is not their primary language, which could have impacted on the perspectives expressed during the interviews.

## Conclusions

This study examines the process of implementing supportive supervision in humanitarian contexts, a much-needed mechanism to protect the wellbeing of MHPSS humanitarian practitioners and preserve the quality and sustainability of their services. Participants delineate the steps and preparations needed for successful implementation and identify the impediments of implementing supportive supervision, highlighting the need for cultural and contextual adaptation, language translation and advocacy to advance an organisation's supervisory structure towards best practice. The IMS was felt by participants to provide a useful framework to guide the implementation of supportive supervision and advocate for supportive supervision within humanitarian organisations in diverse contexts.

## Supporting information

**S1 Checklist.**
(DOCX)

## Acknowledgments

Many thanks to our participants for their time and valuable contributions to our project.

## Author Contributions

**Conceptualization:** Nadeen Abujaber, Meg Ryan, Kelly A. McBride, Michelle Engels, Frédérique Vallières.

**Data curation:** Nadeen Abujaber, Meg Ryan, Frédérique Vallières.

**Formal analysis:** Nadeen Abujaber, Meg Ryan, Frédérique Vallières.

**Funding acquisition:** Frédérique Vallières.

**Investigation:** Nadeen Abujaber, Meg Ryan, Frédérique Vallières.

**Methodology:** Nadeen Abujaber, Meg Ryan, Frédérique Vallières.

**Project administration:** Kelly A. McBride, Pia Tingsted Blum, Michelle Engels, Anna Didenko, Hannah Green, Catia Sofia Peres de Matos, Shona Whitton, Frédérique Vallières.

**Resources:** Nadeen Abujaber, Meg Ryan, Anna Didenko, Hannah Green, Frédérique Vallières.

**Supervision:** Frédérique Vallières.

**Validation:** Meg Ryan, Frédérique Vallières.

**Visualization:** Meg Ryan, Anna Didenko, Hannah Green, Frédérique Vallières.

**Writing – original draft:** Nadeen Abujaber, Meg Ryan, Frédérique Vallières.

**Writing – review & editing:** Nadeen Abujaber, Meg Ryan, Kelly A. McBride, Pia Tingsted Blum, Michelle Engels, Anna Didenko, Hannah Green, Catia Sofia Peres de Matos, Shona Whitton, Frédérique Vallières.

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
