## [Decision Letter · Decision Letter 0]

18 Aug 2023

PGPH-D-23-01248

Implementing Supportive Supervision in Acute Humanitarian Emergencies: Lessons Learned from Afghanistan and Ukraine

Dear Dr. Abujaber,

Thank you for submitting your manuscript to PLOS Global Public Health. After careful consideration, we feel that it has merit but does not fully meet PLOS Global Public Health’s publication criteria as it currently stands. Therefore, we invite you to submit a revised version of the manuscript that addresses the points raised during the review process.

Broadly the article is recognized as being of merit but further clarification is required. Reviewer 1 and Reviewer 2 have both indicated that they strongly support publication. However, each has requested methodological clarifications regarding small sample size, characteristics of respondents, interaction among respondents, and other descriptive details. Please attend to both sets of feedback and provide your revisions or your rebuttals. 

We look forward to receiving your revised manuscript.

Kind regards,

Sharon Alane Abramowitz, Ph.D.

Academic Editor

Journal Requirements:

2. Please include the following request in the decision letter, and ping me with follow-up. “Please include a complete copy of PLOS’ questionnaire on inclusivity in global research in your revised manuscript. Our policy for research in this area aims to improve transparency in the reporting of research performed outside of researchers’ own country or community. The policy applies to researchers who have travelled to a different country to conduct research, research with Indigenous populations or their lands, and research on cultural artefacts. The questionnaire can also be requested at the journal’s discretion for any other submissions, even if these conditions are not met.  Please find more information on the policy and a link to download a blank copy of the questionnaire here: https://journals.plos.org/globalpublichealth/s/best-practices-in-research-reporting. Please upload a completed version of your questionnaire as Supporting Information when you resubmit your manuscript.”

3. In the online submission form, you indicated that "Data can be made available by the author upon reasonable request". All PLOS journals now require all data underlying the findings described in their manuscript to be freely available to other researchers, either 1. In a public repository, 2. Within the manuscript itself, or 3. Uploaded as supplementary information.

Additional Editor Comments (if provided):

Reviewers' comments:

Reviewer's Responses to Questions

**Comments to the Author**

1. Does this manuscript meet PLOS Global Public Health’s publication criteria? Is the manuscript technically sound, and do the data support the conclusions? The manuscript must describe methodologically and ethically rigorous research with conclusions that are appropriately drawn based on the data presented.

Reviewer #1: Yes

Reviewer #2: Yes

2. Has the statistical analysis been performed appropriately and rigorously?

Reviewer #1: N/A

Reviewer #2: N/A

3. Have the authors made all data underlying the findings in their manuscript fully available (please refer to the Data Availability Statement at the start of the manuscript PDF file)?

Reviewer #1: Yes

Reviewer #2: No

4. Is the manuscript presented in an intelligible fashion and written in standard English?

Reviewer #1: Yes

Reviewer #2: Yes

5. Review Comments to the Author

Reviewer #1: This is a nicely written manuscript that highlights an important problem and a relevant intervention. The main limitation is the small sample but this is understandable. Would it be possible to know more about the participants besides role and gender without breaching confidentiality? Years of experience in the field, nationality etc. This would be helpful to contextualize their answers.

Given the importance importance I would have liked more references and emphasis in the discussion and conclusion to the problem itself outside the confines of the organization and the specific intervention. Basically how can other organizations in the humanitarian field learn from it or use it.

Otherwise I believe it is an interesting paper that deserves publication

Reviewer #2: 1. This is an action research and the authors have presented data from the interviews of five participants.

It was however not clear if all five participants were also involved in a group discussion. The authors used the word 'discussed' while presenting the results. See for example lines 214, 236, 253 282. Themes and sub- themes are appropriate and relevant.

2. The study being a qualitative one did not require statistical analysis.

3. The authors mentioned that the availability of the data will be upon request during to the sensitivity of the topic under study

6. PLOS authors have the option to publish the peer review history of their article (what does this mean?). If published, this will include your full peer review and any attached files.

**Do you want your identity to be public for this peer review?** For information about this choice, including consent withdrawal, please see our Privacy Policy.

Reviewer #1: **Yes: **Joseph El-khoury MD MSc FRCPsych

Consultant Psychiatrist

Medical Director

The Valens Clinic

Reviewer #2: No

---

## [Decision Letter · Decision Letter 1]

19 Dec 2023

PGPH-D-23-01248R1

Implementing Supportive Supervision in Acute Humanitarian Emergencies: Lessons Learned from Afghanistan and Ukraine

Dear Dr.Nadeen Abujaber

Thank you for submitting your manuscript to PLOS Global Public Health. After careful consideration, we feel that it has merit but does not fully meet PLOS Global Public Health’s publication criteria as it currently stands. Therefore, we invite you to submit a revised version of the manuscript that addresses the points raised during the review process.

The reviewers have agreed that this paper is technically sound and should be considered for publication. A few small comments to address first.

Reviewer 1

1. There are couple of grammatical errors and would advise the authors to check for all long sentences that are unclear - example - lines 99, 100, lines 104 -108, lines 108 - 112.

Reviewer 2

1. Ethics approval clarification - "It will be interesting to know if the ethical approval from Trinity College Dublin Health Policy and Management /Centre for Global Health ethics committee which is not based in the countries where data was collected, is valid for this study."

2. proofing "There is an omission to italicize a t on line 299 on the reported speech of a participant in the study."

We look forward to receiving your revised manuscript.

Kind regards,

Amy Parry

Academic Editor

Journal Requirements:

Reviewers' comments:

Reviewer's Responses to Questions

**Comments to the Author**

1. If the authors have adequately addressed your comments raised in a previous round of review and you feel that this manuscript is now acceptable for publication, you may indicate that here to bypass the “Comments to the Author” section, enter your conflict of interest statement in the “Confidential to Editor” section, and submit your "Accept" recommendation.

Reviewer #2: All comments have been addressed

Reviewer #3: (No Response)

2. Does this manuscript meet PLOS Global Public Health’s publication criteria? Is the manuscript technically sound, and do the data support the conclusions? The manuscript must describe methodologically and ethically rigorous research with conclusions that are appropriately drawn based on the data presented.

Reviewer #2: Yes

Reviewer #3: Yes

3. Has the statistical analysis been performed appropriately and rigorously?

Reviewer #2: N/A

Reviewer #3: N/A

4. Have the authors made all data underlying the findings in their manuscript fully available (please refer to the Data Availability Statement at the start of the manuscript PDF file)?

Reviewer #2: No

Reviewer #3: Yes

5. Is the manuscript presented in an intelligible fashion and written in standard English?

Reviewer #2: No

Reviewer #3: Yes

6. Review Comments to the Author

Reviewer #2: 1. Availability of data is indicated upon request and the authors have provided satisfactory reason for it.

2. There are couple of grammatical errors and would advise the authors to check for all long sentences that are unclear - example - lines 99, 100, lines 104 -108, lines 108 - 112.

Reviewer #3: The manuscript is technically sound, despite the small sample size and the unrepresentative nature of participants, it provides critical information on the process of planning for the implementation of the IMS, the lessons learned, and the challenges in the implementation of the IMS in 2 acute humanitarian contexts. It would have been interesting to get the feedback of supervisees in this study. The data management procedure is scientifically sound. The data transcription, thematic analysis using the Braun and Clarks framework, and the creation of sub-themes and direct quotations are excellent. In addition, the data can be made available upon request. It will be interesting to know if the ethical approval from Trinity College Dublin Health Policy and Management /Centre for Global Health ethics committee which is not based in the countries where data was collected, is valid for this study.

The manuscript is presented in an intelligible fashion, in plain and easy-to-understand English language. There is an omission to italicize a t on line 299 on the reported speech of a participant in the study. Another advocacy point for the implementation of the supportive suppervission during acute humanitarian emergencies is to involve the coordination mechanisms put in place for MHPSS during a humanitarian crisis to adopt the framework as a means of building the capacities and supporting the members of the MHPSS technical working group.

7. PLOS authors have the option to publish the peer review history of their article (what does this mean?). If published, this will include your full peer review and any attached files.

**Do you want your identity to be public for this peer review?** For information about this choice, including consent withdrawal, please see our Privacy Policy.

Reviewer #2: No

Reviewer #3: **Yes: **Chandini Aliyou Moustapha

---

## [Decision Letter · Decision Letter 2]

15 Mar 2024

Implementing Supportive Supervision in Acute Humanitarian Emergencies: Lessons Learned from Afghanistan and Ukraine

PGPH-D-23-01248R2

Dear Dr Abujaber,

We are pleased to inform you that your manuscript 'Implementing Supportive Supervision in Acute Humanitarian Emergencies: Lessons Learned from Afghanistan and Ukraine' has been provisionally accepted for publication in PLOS Global Public Health.

Best regards,

Saloni Dev

Academic Editor

Reviewer Comments (if any, and for reference):

Reviewer's Responses to Questions

**Comments to the Author**

1. If the authors have adequately addressed your comments raised in a previous round of review and you feel that this manuscript is now acceptable for publication, you may indicate that here to bypass the “Comments to the Author” section, enter your conflict of interest statement in the “Confidential to Editor” section, and submit your "Accept" recommendation.

Reviewer #2: All comments have been addressed

Reviewer #3: All comments have been addressed

2. Does this manuscript meet PLOS Global Public Health’s publication criteria? Is the manuscript technically sound, and do the data support the conclusions? The manuscript must describe methodologically and ethically rigorous research with conclusions that are appropriately drawn based on the data presented.

Reviewer #2: Yes

Reviewer #3: Yes

3. Has the statistical analysis been performed appropriately and rigorously?

Reviewer #2: Yes

Reviewer #3: Yes

4. Have the authors made all data underlying the findings in their manuscript fully available (please refer to the Data Availability Statement at the start of the manuscript PDF file)?

Reviewer #2: Yes

Reviewer #3: Yes

5. Is the manuscript presented in an intelligible fashion and written in standard English?

Reviewer #2: Yes

Reviewer #3: Yes

6. Review Comments to the Author

Reviewer #2: There are some minor grammatical corrections to be done:

1. Under Implications - lines 602-608 is a long one sentence paragraph. Authors can consider breaking it into 2-3 good sentences. 2. Grammatical error in lines 578-579, the sentence is beginning with the word 'which'.

Authors have declared the availability of data upon reasonable request.

Reviewer #3: 1. All the comments were addressed except the comments that had to do with having a uniform style for the reported speech on lines 254, 300, and 329. The use of italics

2. From my previous review, the manuscript was technically sound

3. The Statistical analysis has been performed rigorously.

4. The data can be made available on request.

5. The manuscript is presented in an intelligle fashion in standard English

7. PLOS authors have the option to publish the peer review history of their article (what does this mean?). If published, this will include your full peer review and any attached files.

**Do you want your identity to be public for this peer review?** For information about this choice, including consent withdrawal, please see our Privacy Policy.

Reviewer #2: **Yes: **Susan Julia Chand

Reviewer #3: **Yes: **Chandini Aliyou Moustapha
